

# Guidelines for using empirical studies in software engineering education

Fabian Fagerholm[1,*], Marco Kuhrmann[2,*] and Jürgen Münch[1,3,*]

[1] Department of Computer Science, University of Helsinki, Helsinki, Finland
[2] Institute for Applied Software Systems Engineering, Clausthal University of Technology, Goslar, Germany
[3] Herman Hollerith Center (HHZ), Reutlingen University, Böblingen, Germany
[*] These authors contributed equally to this work.

## ABSTRACT

Software engineering education is under constant pressure to provide students with industry-relevant knowledge and skills. Educators must address issues beyond exercises and theories that can be directly rehearsed in small settings. Industry training has similar requirements of relevance as companies seek to keep their workforce up to date with technological advances. Real-life software development often deals with large, software-intensive systems and is influenced by the complex effects of teamwork and distributed software development, which are hard to demonstrate in an educational environment. A way to experience such effects and to increase the relevance of software engineering education is to apply empirical studies in teaching. In this paper, we show how different types of empirical studies can be used for educational purposes in software engineering. We give examples illustrating how to utilize empirical studies, discuss challenges, and derive an initial guideline that supports teachers to include empirical studies in software engineering courses. Furthermore, we give examples that show how empirical studies contribute to high-quality learning outcomes, to student motivation, and to the awareness of the advantages of applying software engineering principles. Having awareness, experience, and understanding of the actions required, students are more likely to apply such principles under real-life constraints in their working life.

Corresponding author
Fabian Fagerholm,
fabian.fagerholm@helsinki.fi

## INTRODUCTION

Providing relevant knowledge and skills is a continuous concern in software engineering education. Students must be exposed to realistic settings to understand why applying fundamental software engineering principles is necessary, why decisions should be grounded in evidence, and to learn to foresee long-term and delayed effects of certain behaviour or decisions in software projects. Using empirical instruments is one approach to teach relevant software engineering knowledge and skills. The goal of this paper is to use our teaching experiences to develop practice-grounded guidelines that help teachers include empirical instruments in their teaching.

Since real-life software development routinely deals with large, software-intensive systems and is influenced by the manifold and complex effects of teamwork and distributed

software development, software engineering education must enable students to understand such environments and to apply knowledge properly and effectively. However, restrictions in the academic curriculum and the complexity and criticality of real software products limit the level of realism that can be achieved in education. As problems are narrowed down to be manageable, practical relevance is lost through scope and problem size limitations and the use of artificial settings instead of real-world problems. Many effects only become visible over long time periods, e.g., the efficiency of a particular method or eventual impact of a design decision. Often, time is too limited to provide adequate means to experience such effects in a single course. The same problem occurs in practitioner training. Industry must quickly develop solutions and services in order to deliver customer value and, eventually, survive in market competition. Empirical evidence may not be easily available, and practitioners may resort to decision-making based on by biased individual beliefs, negatively affecting the productivity of development teams.

Over the years, we have implemented empirical instruments in software engineering courses to (1) provide an environment in which students can experience real-life problems while increasing their motivation and the quality of learning outcomes, (2) pave the way for conducting research in collaboration with industry, and (3) apply these instruments in industry for training purposes. In our experience, this approach has been well suited to prepare students for working life. Training in empirical instruments, such as experimentation or case study research, and direct experience of the value provided by them, has encouraged our students to apply their acquired knowledge in practice, and to explore problems, new methods, and new tools in a systematic and evidence-based manner. Since empirical instruments are well accepted for conducting (applied) research in industry, and since students can form their own experiences when doing empirical studies, we claim that utilizing empirical instruments in teaching increases the quality as well as the practical relevance of SE education.

We show how different instruments, ranging from controlled experiments to qualitative studies, can be used for teaching purposes. We also consider overarching approaches that situate empirical studies in a larger context. We systematize purposes and challenges of the different study types, discuss the validity of the results that can be obtained in a teaching context, and create a link between teaching and research. We use a selection of representative studies to discuss the impact on teaching as well as on industry relevance. The guidelines developed in this paper provide a systematic collection of purposes, learning goals, challenges, and validity constraints, and aims to support teachers in selecting proper study types for inclusion into their courses.

The remainder of this paper is structured as follows. 'Related Work' reviews related work on empirical studies as a teaching instrument. 'Research Approach' describes the research approach taken in this paper. 'An Overview of Empirical Study Types for Software Engineering Education' discusses empirical instruments in SE education and provides an initial taxonomy of them. 'A Guideline for Integrating Empirical Studies with Software Engineering Courses' presents the main contribution of the paper: a guideline for integrating empirical studies with software engineering courses. 'Experiences' gives examples of implementing empirical instruments in university education and discusses the

implications of integrating them into courses. 'Conclusion' provides conclusions and lists possible future work.

## RELATED WORK

Experiments and other types of empirical studies are key to the scientific approach. Empirical studies are performed in research for many different purposes, such as understanding real-world phenomena, testing hypotheses, and validating theories (*Wohlin et al., 2012*; *Runeson et al., 2012*; *Kitchenham, Budgen & Brereton, 2015*). Empirical studies, especially experiments, are established only in few disciplines. The use of empirical studies as teaching tools is less common than, for example, classroom exercises and lectures. In many areas, such as software engineering, such utilization of empirical studies is still uncommon compared to learning tasks that involve reading or writing about existing research, and individual exercises that focus mainly on small-scale technical implementation.

### Empirical studies in teaching in other disciplines

Physics education may be a prime example where, due to the historical development of the discipline, experiments have a central pedagogical role. Beyond their function as means for verifying or refuting theories, experiments in physics have a generative function with relevance for education and learning (*Koponen & Mäntylä, 2006*). While the type of problems in physics and software engineering are different, experiments play a similar role for learning in both. An example of another discipline, also different in nature from physics, that already has a high level of maturity in using experiments for teaching purposes is economics. Experiments became widespread teaching tools in economics in the 1990s (*Parker, 2014*). Nowadays, many economists use experiments as educational tools. *Parker (2014)* mentions several benefits of using experiments: they are distinctive and more participative, and in consequence, students are likely to remember lessons associated with them. Parker also mentions that the experiential component in experiments can be very important and that students and instructors usually think that experiments are fun.

Experiments can be used as part of many educational approaches. For example, experiments could be used in different ways with problem-based learning (PBL) (*Barrows & Tamblyn, 1980*; *Wood, 2003*). In PBL, students define their own learning objectives connected to a problem scenario, while the tutor ensures that the objectives are "focused, achievable, comprehensive, and appropriate" (*Wood, 2003*). One possibility is to have the tutor guide students towards objectives that involve different degrees of experimentation, e.g., formulating research questions, defining research designs, or even carrying out studies with real or simulated data. The tutor may provide data as part of the problem scenario; it may be part of the trigger material provided to students. Experiments can also be used as part of project-based learning (*Blumenfeld et al., 1991*), where students actively explore real-world challenges and problems. Instructors can introduce experiments when important decision-making and knowledge acquisition needs emerge.

In order to support the design of constructive education with experiments embedded, as well as to support experimentation within more traditional teaching, teachers would benefit from guidelines or sets of ready-made experiment templates that they could use

either when planning or dynamically during teaching. The SERC Portal for Pedagogy in Action, created by *Ball et al. (2012)*, provides a repository of classroom experiments. Ball et al. define such experiments as ''activities where any number of students work in groups on carefully designed guided inquiry questions''. Students collect data through interaction with typical laboratory materials, data simulation tools, or a decision-making environment, as well as ''a series of questions that lead to discovery-based learning.'' The repository includes a comprehensive list of experiments from different disciplines that can be used for replication in classroom settings. In addition, it contains references to scientific studies that provide empirical evidence about the expected positive effects of experiments as teaching tools. An example is an empirical investigation of the impact of classroom experiments on the learning of economics (*Frank, 1997*). Several of the cited studies show a higher academic achievement (e.g., measured as increase in students' homework scores) when using experiments compared to control classes where standard lectures are used. *Ball et al. (2012)* also cite studies that show improved student satisfaction with teaching pedagogy when using experiments. The repository also contains guidelines for designing and conducting experiments as part of teaching. The guidelines include important aspects such as strategies for unexpected outcomes of experiments.

## Requirements for applying empirical studies in teaching

The discussion regarding the suitability of experiments mainly focuses on criteria that need to be fulfilled for designing successful experiments, the balance between practical work and theory, and the suitability of students as experimental subjects. *Parker (2014)* mentions basically three criteria: (1) the experiment must be aligned with the central topic of the course, (2) the concept to be taught through the experiment should not be easily understood without the experiment or already be obvious, (3) students need to be able to quickly learn the necessary prerequisites for participating in the experiment.

*Dillon (2008)* provides an overview of advantages and disadvantages of experiments based on empirical findings. An important conclusion drawn from the overview is that successful observation of a phenomenon as part of an empirical study should not be an end in itself. Rather, students should have enough time to get familiar with the related ideas and concepts associated with the phenomenon.

## Empirical studies in software engineering education

In software engineering, experimentation was established in the 1980s. *Basili, Selby & Hutchens (1986)* were among the first to present a framework and process for experimentation. Since then, software engineering experiments in classroom settings have become more common. However, the focus of most of such experiments has been to gain research knowledge, with students participating as research subjects. Less attention has been paid to using empirical studies with an educational purpose in mind, where the experiment has an explicit didactic or experiential role. Few curricula are available that include the execution of empirical studies as an integral part of a lecture (e.g., *Kuhrmann, Fernández & Münch, 2013*; *Hayes, 2002*).

The use of students as experimental subjects has often been discussed in the literature. In software engineering, the topic has mainly been analysed for understanding the suitability

of students as subjects compared to professional practitioners as subjects. An example of such an investigation has been presented by *Runeson (2003)*. *Carver et al. (2003)* note that while it is common to carry out empirical studies in software engineering with students as subjects, the educational value of the studies is often overlooked. Simultaneously, solving the pedagogical challenges involved is not straightforward. Carver et al. discuss costs and benefits for researchers, students, instructors, and industry, and provide a check-list with advice for carrying out empirical studies with student subjects. The same authors have later extended their check-list with requirements for successful empirical studies with students, based on previous literature (*Carver et al., 2010*). The check-list includes items addressing considerations before a class begins, as soon as it begins, when the study begins, and when the study is completed. The authors emphasise integration of the study and course topic and schedule, documentation, and considerations of study validity.

Only few studies have investigated the impact on learning of empirical studies in the curriculum. We expect that the effect is generally positive as long as the integration is carried out properly. *Staron (2007)* finds that students' learning process is improved and that including carefully designed experiments into software engineering courses increases their motivation. A large majority (91%) of students who participated as subjects in the experiments found them useful, and the number of high-passes increased by 41% after introducing experiments.

While many articles report on empirical studies using student subjects, and some articles report on the educational benefits of such studies for students, few papers address empirical studies as an overall strategy for software engineering education. In particular, there is a lack of guidance for using empirical studies in software engineering education in cases where students may not only be research subjects but could also be involved in carrying out the studies. An overview that discusses different types of empirical studies, their suitability for education, as well as challenges with respect to their execution is missing.

## RESEARCH APPROACH

The goal of this paper is to develop guidelines that help teachers integrate empirical instruments in software engineering education. The guidelines are based on a reflective analysis of our experiences with teaching courses that use empirical elements to support learning objectives. A reflective approach has been recognised by many educational researchers as a prerequisite for effective teaching (e.g., *Hatton & Smith, 1995*; *Cochran-Smith, 2003*; *Jones & Jones, 2013*). Reflective practice, with roots in the works of *Dewey (1935)* and *Schön (1983)*, calls for continuous learning through deliberate reflection in and on action. Using empirical instruments in software engineering education is a way to encourage students to reflect, but teachers should do the same. This paper represents one outcome of reflection-on-action: we analyse materials, assignments, notes, course syllabi, schedules and structures, evaluation data, and recollections of important factors in a number of our own courses, and derive guidelines that we believe would help teachers implement similar courses.

Our approach is mainly qualitative and has proceeded from gathering a list of study types through analysis of materials and experiences relevant to each study type to the guideline

proposed in this paper. Here, analysis refers to categorisation of materials and identification of connections and relationships between categories. Our main goal of developing the guideline helped to scope our investigation, and we thus left out material which did not serve this goal. We began by sifting through parts of the published literature on software engineering education and methods in order to shape a first outline of a taxonomy of study types. In particular, we were influenced by *Höst (2002)* and *Carver et al. (2003)* when considering software engineering education, and by *Shull, Singer & Sjøberg (2008)* and *Kitchenham, Budgen & Brereton (2015)* when considering the methodological aspects. Our search was purposive rather than systematic, as we sought to construct a taxonomy (see 'An Overview of Empirical Study Types for Software Engineering Education') for use in the guidelines rather than for the purpose of representing the state of the art in the scientific literature.

After constructing the taxonomy, we analysed qualitative data from our own courses and arranged it according to five categories: (1) learning goals, purposes, challenges, and validity, (2) establishing context and goals, and determining a study type, (3) motivating students, (4) scheduling, (5) other considerations. We summarised the qualitative data in each category by removing the details specific to our courses and generalising the insights so that they can be applied more broadly. We then constructed the guideline by cross-referencing the categories so that the purpose, challenges, and validity concerns relevant for each study type is shown. The result is given in 'A Guideline for Integrating Empirical Studies with Software Engineering Courses'.

Finally, we revisited the material from our courses and picked examples that illustrate how we tackled some of the choices teachers face when using empirical instruments for education. We also addressed the specific question of evaluating our teaching by providing data from formal as well as informal evaluation (see 'Experiences'). This serves as a first validation of the guidelines.

# AN OVERVIEW OF EMPIRICAL STUDY TYPES FOR SOFTWARE ENGINEERING EDUCATION

The software engineering literature includes a number of empirical studies with students, and often these studies were conducted in an educational setting. In this section, we give an overview of (empirical) study types utilised in software engineering education. We list common instruments from empirical software engineering and provide examples of how these instruments can be applied to teaching. The overall goal of this section is to summarise different study types that can be used in software engineering education. The summary supports the development of an initial common taxonomy that categorises study types. The taxonomy helps to determine the appropriateness of a particular instrument in a specific setting. A concise overview of the study types, including a brief description and outline of the potential positive educational aspects, is given in Table 1.

## Case studies

Case studies aim to investigate a phenomenon in its natural context. When utilised for educational purposes, case studies can omit some aspects of a full research design (*Yin,*

**Table 1  Summary of empirical study types.**

| Type | Description | Potential for education |
|---|---|---|
| Case study | Investigate phenomenon in its natural context. Especially suitable for exploratory and explanatory designs. Results grounded in context. | Gaining observational and analytic skills. Observing real scenarios with real objectives and constraints. Knowledge of high relevance for professional use. |
| Formal and semi-formal experiment | Investigate effect of treatment under controlled conditions. Rigorous design requirements. Results constitute tests of a theory. | Demonstrate the real impact of theory. Gain skills to formulate and test a theory. |
| Continuous experimentation | Constant series of experiments to test value creation, delivery, and capture of software or software-based products. Results of experiments can be used to make design decisions. | Understand connection between software development and business and customer domain. Gain skills to test product assumptions to provide evidence for product decisions. |
| Software process simulation | Simulation model used as abstraction of a real process. Cost and time advantages can be obtained. Requires a valid model. | Gain understanding of process dynamics and complexity with limited resources. Experience effects of decisions. |
| Individual studies | E.g., Bachelor's or Master's theses. Focused work on a specific problem for a limited time. Various studies are possible. | Learn to conduct a study in a self-organised manner. Gain domain knowledge. |
| Further instruments | Augment or provide context for aforementioned study types, e.g., replication studies, OSS projects. | Provide ways to enhance other study types. |

*2009*), but can borrow from design science methodology (*Hevner et al., 2004*) where an artefact is designed, implemented, and evaluated in order to learn.

When performing case studies with industry, the context is provided by business objectives and realistic constraints (*Brügge, Krusche & Alperowitz, 2015*). Industry aims to develop results which contribute to solving a problem with relevance in their settings. For instance, developers can be trained in a close-to-reality environment, aiding the understanding of situations that will occur in the (near) future. On the other hand, researchers perform case studies to understand and capture phenomena in their natural context. Depending on the rigorousness of the study design, both practitioners and researchers benefit from a case study due to its grounding in realistic settings. Apart from "normal" case study research, teachers can use the case study instrument to help motivate students by providing problems with visible real-life applications. Case studies also help teachers to transmit procedural knowledge, as students are required to formulate problems and design solutions, and to evaluate them.

Case studies help answering explanatory questions of the type "How?" or "Why?" They should be based on an articulated theory regarding the phenomenon of interest (*Yin, 2009*). A case study can then provide additional evidence for a theory, help to modify or refine a theory, or suggest an alternative theory that better fits the observations. Furthermore, a case study can also be utilized to discover (new) interesting and relevant issues. Case studies can be implemented in different ways. They can be categorized as single- or multiple-case, holistic or embedded (*Wohlin et al., 2012*; *Runeson et al., 2012*; *Yin, 2009*), or as intrinsic (*Stake, 1995*; *Baxter & Jack, 2008*). They can be deductive or inductive, exploratory or

confirmatory, and they can make use of both quantitative and qualitative data (*Yin, 2009*; *Eisenhardt, 1989*). In the context of teaching, the normal case study setup is a holistic single-case study in which a single instance of the unit of analysis (case) is examined. More complex designs, e.g., multiple-case studies, increase the value of the study results for research. However, these aspects can be considered less important for teaching. Furthermore, setting up a case study—even in teaching—requires an environment in which the phenomenon of interest occurs naturally.

Case studies are a valuable source for generating diverse results. From the industry point of view, case studies help to elaborate and understand the value given by reaching the case objective, ranging from increased technological understanding to increased understanding of customer value of a product or service. They help uncover real technology- and knowledge-related challenges involved in reaching the objective and, thus, provide information on the cost, effort, and risks involved in the case. From the perspective of researchers, case studies can contribute to the development of general—but context-bound—technological rules, including case-specific insights and lessons learnt. Given replication with multiple cases, the rules can also reach the level of more general theory. Moreover, case studies can be fruitful grounds for exploration and help to discover or identify research questions.

From the teaching perspective, case studies serve several purposes of which gaining observational and analytic skills are the most important. Case studies help participants to get insights into a setting in which a particular phenomenon occurs. Therefore, analysing problems and deriving tasks to solve the problem happens in real scenarios rather than synthetic situations. Solutions can be evaluated against real objectives and constraints. Consequently, this kind of learning produces knowledge of higher relevance for professional use, and teaching directly addresses subject matter and procedural knowledge related to a specific problem type.

### *Examples of case studies in teaching*

*Fagerholm, Oza & Münch (2013)* describe the Software Factory, which is an instrument to combine software development education and training with conducting empirical research. A fruitful ground for this kind of teaching is global software development, as demonstrated by, e.g., *Oza et al. (2013)*, *Richardson, Milewski & Mullick (2006)*, and *Deiters et al. (2011)*. In the Software Factory environment, students work with a company on a real software development project, providing a level of realism that is not available in a regular course exercise. This realism, along with the opportunity to work in a team setting provides the potential to conduct case studies with educational relevance.

### Formal and semi-formal experiments

According to *Wohlin et al. (2012)*, an experiment (controlled experiment) is defined as "an empirical inquiry that manipulates one factor or variable of the studied setting." Different treatments are applied, or treatments are assigned to different subjects, to measure effects on variables of interest. If treatment is not randomly assigned, we speak of a "quasi-experiment." Experiments aim to investigate settings in which the environment

is under control, and effects of interest are investigated by manipulating variables. For instance, if the efficiency of a particular method is subject to investigation, one experiment group is assigned to solve a problem with the "new" method, while another group works on the same task, but using another method. Results are then compared, e.g., to accept or reject a hypothesis. Thus, experiments can be utilised to test theories and conventional wisdom, explore relationships, to evaluate the accuracy of models, and to validate measures. Importantly, experiments should always provide a detailed context description, showing the settings in which certain claims are true, and in which certain techniques or tools are beneficial.

Experiments require rigorous design. *Wohlin et al. (2012)* present an experiment process which consists of scoping, planning, operation, analysis and interpretation, and presentation and packaging. However, providing a general experiment design is demanding, as the design depends on the respective subject and context. Apart from the general experiment process, several smaller guidelines exist to direct researchers through the process, e.g., the goal template from the TAME project (*Basili & Rombach, 1988*), experimentation packages providing reusable designs, templates, and so forth (e.g., in the context of self-organizing project teams (*Kuhrmann & Münch, 2016b*), we created such a template; another example can be found in *Fucci, Turhan & Oivo (2015)*), and, pragmatically, case study designs, which can be derived from, e.g., *Runeson & Höst (2009)*, who provide advice on planning and a guideline on reporting case study research.

Experiments provide several trade-offs for the conducting parties. However, the usefulness depends on the respective context. For instance, while the importance of experiments in research is not questioned, experimentation in industry has to be considered in terms of business value (e.g., by providing new, efficient methods or creating and evaluating software prototypes, paving the way for new products). Requirements regarding the validity of the results differ as well as the general scope of experiments. Furthermore, as we have previously discussed, small and very small companies usually have insufficient resources to invest in the necessary preparation and allocation of resources (*Kuhrmann, 2015*). Nevertheless, experimentation allows for, e.g., evaluating different methods and tools, building a hypothesis, and testing the hypothesis. Furthermore, experiments help to confirm conventional wisdom, e.g.: "Everybody says that Follow-the-Sun development is fast but expensive—is this also true for our situation?".

### Examples of experimentation in teaching

For teaching, experiments can be a valuable source for knowledge and experience. For instance, experiments can be used to elaborate the real impact of theoretical concepts: in *Kuhrmann & Münch (2016b)*, the theoretical concept taught was the well-known Tuckman Model (*Tuckman, 1965*), and in a complementing experiment, students could experience the effects of group dynamics themselves, e.g., changing team set-ups or external influences. Another experiment reported in *Kuhrmann & Münch (2016a)* creates a setting in which students can experience the crucial role of communication (and absent communication) in distributed development set-ups.

In *Kuhrmann, Fernandez & Knapp (2013)*, we present a controlled experiment on the perception of software process modelling paradigms. A German NGO sponsored a process description, which the students had to analyse and improve according to a given approach (*Kuhrmann, Fernández & Münch, 2013*). Students used two different process development environments, each implementing a different modelling paradigm. They went through the process life cycle, learned about analysis-, design-, and realisation tasks, and conducted result assessments. Furthermore, the experiment outcomes showed advantages and disadvantages of the particular modelling paradigms.

## Continuous experimentation

Continuous experimentation refers to a constant series of experiments to test the value of software capabilities such as features early in its design process (*Fagerholm et al., 2014a*; *Fagerholm et al., 2017*). The major driver is the industrial need to better understand product value delivery, so that development activities can be focused on delivering only capabilities that create value for users or customers. Our experience shows that it is a mistake to ignore the value aspect in SE education, as it is a critical part of understanding software requirements. This is especially relevant in complex domains where requirements are unknown and cannot be elicited up front.

In contrast to empirical software engineering that usually focuses on technical product or process aspects from a developer perspective, the purpose of continuous experimentation is to validate assumptions that are underlying a business model or a product roadmap. The perspective is usually that of a product owner or entrepreneur. Continuous experimentation is a means to evolve business models, product roadmaps, or feature scopes based on validated assumptions. It is based on approaches such as Lean Startup (*Ries, 2011*) and Customer Development (*Blank, 2006*), and enforces product managers and developers to connect with real users (e.g. through interviews or by analysing usage data) in order to test critical assumptions and make evidence-based product decisions. This typically requires, e.g., the execution of experiments in a scientific style and the implementation of feedback channels that allow observing user behaviour. For teachers, it is a great challenge to instruct students accordingly, as classic software engineering teaching is usually separated from value considerations. Thus, creating a mind-set in which value creation is the baseline for all development tasks impacts the way software development is performed in the sense that decision-making is based on continuously obtained evidence about customer value.

In order to conduct continuous experimentation, different designs can be applied depending on the hypotheses or study goals under investigation. A typical design is a case study consisting of a sequence of build-measure-learn cycles that develop a so-called minimum viable product (MVP). Simply speaking, an MVP is a prototype that allows for testing with potential customers. Such testing requires a design to quickly obtain customer feedback during the study. In consequence, access to potential customers is needed to conduct such studies.

From the industry perspective, continuous experimentation results in knowledge that supports or refutes assumptions about product value. An MVP might result from an experiment. Such a result might consist of a working prototype as well as data or

lessons learnt about the customer value of the prototype, its development process, and potentially about relevant customer segments. The study might also contribute to testing of other critical assumptions of a business model, such as assumptions about customer relationships or channels.

For researchers, continuous experimentation helps to better understand processes, methods, techniques, tools, and organizational constraints regarding building the "right" software.

From the teaching perspective, continuous experimentation helps students understand the connection between software development techniques and business. Since such experiments must begin by analysing product-related assumptions, students naturally come into contact with the product's business model. They must then make the link between such assumptions and the corresponding technical implementation and devise an experiment which allows them to refute or support the highest-priority assumption, yielding evidence for a product-related decision. Continuous experimentation can thus foster the awareness of relevant criteria for software beyond cost, reliability, and effort, e.g., usability, usefulness, success (e.g., contribution to a higher level organizational goal), and scalability (e.g., monetization from a significant amount of users).

### Examples of continuous experimentation in teaching

*Fagerholm et al. (2014a)* present building blocks for continuous experimentation and describe the execution of three student projects that aim at conducting build-measure-learn loops. These projects were performed in cooperation with a start-up and sought to understand aspects such as future development options or scalability issues of a new service. The projects helped to evolve the product roadmap and led to several technical pivots where previous assumptions were invalidated and new options found. Students gained significant insights in the connections between technical and business considerations. A process model and infrastructure architecture model for continuous experimentation are described in *Fagerholm et al. (2017)*.

*Kohavi et al. (2012)* describe continuous experiments from an industry perspective. The authors present a system for constant experimentation at Microsoft. They emphasize that learning addresses many aspects beyond understanding experimentation techniques. For instance, it is necessary to learn how to identify and understand the reasons for experiments in an organization. In addition, learning needs to address a change of the company culture towards experimentation.

## Software process simulation

Experimentation is a costly way to learn. It requires, for instance, significant preparation of experimental materials and treatments. Software process simulation refers to the use of a simulation model as an abstraction of a real process. Typical purposes for using such models are experimentation, increased understanding, prediction, decision support, or education about a process. Assuming that a valid model exists, process simulation promises advantages with respect to cost. Since part of the process can be conducted virtually, the number of controlled variables can be much higher than in real experiments, and calibrating the model to a specific context can be done efficiently.

Simulation may be a suitable teaching aid in many situations, but should be used only when a valid model can be obtained. Otherwise, there is a risk that students observe effects that are not realistic and thus incorrect learning might occur. Well-researched models with extensive validation are necessary. Other disciplines such as mechanical engineering or molecular chemistry already use simulation to analyse technologies and processes and thereby reduce the need for real experiments. In software engineering, this trend is still focused towards product aspects such as understanding the dynamic behaviour of control software. However, simulation has already been applied successfully for understanding and analysing software processes as well as for educational purposes. Process simulation can be combined with real software engineering experiments—for example, by using empirical data to calibrate a model or by comparing such data with simulation results—or used as such.

Simulation-based experiments can be classified by the number of treatments and the number of subject groups per treatment. In case of single project studies (i.e., one treatment and one group), simulation requires initialization of appropriate input parameters and calibration to the context. In case of multi-project variation (i.e., more than one treatment and one group), the simulation model needs to be calibrated to different contexts. Replications (i.e., one treatment but more than one group) basically refer to several simulation runs, typically with statistically based variations. In case of blocked subject-project studies (i.e., more than one treatment and more than one team) simulation model development requires a good understanding of cause–effect relations in varying contexts. Combining simulation with experiments can be done in the following ways:

- Empirical knowledge from real experiments can be used for creating the simulation model (e.g., to calibrate the simulator)
- Results from simulation runs can be used for designing real experiments (e.g., to identify and investigate new hypotheses before performing expensive real experiments)
- Both can be done in parallel (e.g., to broaden the scope of the experiment).

From the research perspective, software process simulation can be seen as an additional, efficient mechanism to gain knowledge about the effects of processes in different contexts. It especially allows for analysing situations that are difficult, expensive, or impossible to analyse in real experiments, and it allows for flexible variations of the context and the controlled variables.

From the educational perspective, simulation helps to gain a better understanding of the dynamics of software development processes, getting immediate feedback, and experiencing the effects of decisions. Feedback can be obtained quickly using time-lapse effects. When creating the model, students gain insights into cause–effect- and other relationships. Creating a simulation model promises to improve understanding of the key factors and complexity of a specific software process.

From an industry perspective, learning cycles can be accelerated, risks mitigated, and the impact of processes, technologies, and their changes can be better understood.

### *Examples of software process simulation in teaching*

Several educational software engineering simulation environments have been developed and are used for teaching purposes. Examples are the comprehensively evaluated SimSE environment (*Navarro & Van der Hoek, 2007*), and the SESAM (Software Engineering Simulation by Animated Models) environment (*Ludewig et al., 1992*). *Münch, Rombach & Rus (2003)* have developed a laboratory that allows to systematically combine real and virtual experiments and demonstrate the benefits of such a combination for teaching purposes.

## Individual studies

All aforementioned study types allow for teamwork and training specific team-related skills. However, software engineering education also comprises several individual tasks, which are often performed by students while they work (for a limited time) in industry or while they write their theses (e.g., Bachelor's or Master's thesis, or semester projects). Although individual studies can be performed in industry-academia collaborations, they are usually conducted by individual students who work on a specific task and simultaneously perform the study. The student is thus a participant-observer in such studies.

Individual studies depend on the setting in which they are carried out, e.g., requirements for a semester project differ from those of a Master's thesis. Different study types can be applied. Individual studies have high requirements regarding the study design, as they all have limited resources and strict time constraints in common. Specific challenges are scoping the study, narrowing down research questions, and defining the expected outcome. Since individual students conduct single studies, results are often limited to proofs of concept or demonstrators. Finally, data generated in this kind of study is often isolated, requiring a defined context to which it can contribute, e.g., a more comprehensive research strategy within which a particular study investigates one small aspect.

Although limited, the results of the study can have inherent value to research and practice. For research, the study may contribute to a better understanding of research questions and might be a starting point for further, more comprehensive studies. From the industry perspective, individual studies allow investigating a specific problem in a well-defined environment and, due to their limitations, analyses are limited to a specific problem. For instance, if the objective of the study is to develop an algorithm or to examine the feasibility of a specific method, a case study can be conducted explicitly focusing on this aspect, resulting in a statement which then provides rationale for further investigation. If the individual study is combined with an internship, students can summarise existing research on a topical area that can then be used as training material for company employees. Finally, when students graduate, companies may wish to employ them (they already know the student).

From an educational perspective, students have to work in a self-organised manner, and they learn how to set up and conduct a problem-oriented study (including all effects, e.g., stakeholder interaction, study planning, data collection, etc.). Furthermore, students get further specific domain knowledge beyond the more general knowledge they acquire at university.

### Examples of individual studies in teaching

*Rein & Münch (2013)* describe an individual student study that was performed as part of a seminar thesis. The study aims at analysing features of a mobile app and consisted of design, instrumentation of the app with appropriate measurement instruments, and analysis of data from more than ten thousand users. The study results provided valuable, data-based justifications on how to further develop the app. In addition, a new method for analysing feature value was piloted and experiences with the applicability of the method were gained.

A popular example for individual studies in industry is the so-called Personal Software Process (PSP), a training that consists of a series of systematically defined software engineering exercises. *Rombach et al. (2008)* have analysed data from 3,090 engineers conducting the PSP. A major finding from this analysis is that the effects of applying software engineering principles can be experienced on an individual level. Although the effects of applying such principles typically can only been seen on the larger scale (e.g., large projects, long-lasting development efforts, multi-team developments), this study shows that it is possible to teach these principles also on the individual level.

## Further instruments

In the previous sections, we provided an overview of different empirical instruments, a discussion, and examples of their application in teaching. However, there are further means that can contribute to the aforementioned study types.

### Replication studies

Replication provides an opportunity to learn from an already established research design, and can, if conducted well, contribute additional evidence for a research question. Replication repeats empirical studies to solidify their results, test result reproducibility, increase result validity (e.g., *Easterbrook et al. (2008)* and *Park (2004)* consider replication a kind of triangulation), and broaden research context and scope by repetition under similar conditions while changing selected variables, e.g., site, population, and instruments. Thus, students can learn from adapting the research design in a new environment and by comparing the results obtained to those of the original study. Simultaneously, teachers should prepare students for sometimes large differences in results. Lack of generalisability is often cited as a limitation in empirical studies and replication is a step toward creating generalisable knowledge. However, replication in software engineering is considered immature and is subject to debate. *Juristo & Gómez (2012)* argue that results from current software engineering experiments are often produced by chance, are artificial, and are too far away from reality. They mention that the key experimental conditions are yet unknown, as the tiniest change in study design may lead to inexplicable results. Due to the large number of varying factors in the context of software engineering, it can be questioned whether close replication is possible at all (*Juristo & Vegas, 2011*). Nevertheless, conducting a replication can be a valuable learning experience which develop students' ability to design studies and critically compare results of studies addressing the same or similar questions.

Although industry-based research is considered the optimal way to gather reliable and relevant data, empirical research in industry is hard; replications are even harder.

For instance, as we discussed in *Kuhrmann (2015)*, small companies usually have limited resources to conduct empirical research, as it requires preparation, time, and allocation of resources. *Armbrust et al. (2008)* mention the importance of pilot projects in the context of case study research, and discuss the difficulties finding proper projects and allocating resources. Replication increases the level of difficulty, as experiments and case studies are conducted multiple times thus blocking critical resources for long periods of time without immediately creating value in terms of products and services. While replication in industry is hard to implement, replicated experiments and case studies are easier to realize in education since universities provide a stable environment. Deviations from original designs are usually the subjects, while, e.g., case, instruments, and procedures can be kept stable. Thus, once a consolidated experiment design is in place, replications can be implemented on a regular basis. However, the question of whether results obtained with student subjects can be generalised to industry is then of crucial importance.

### Real-life examples and open source software

A major challenge for teachers is providing students with problems of considerable size. One approach is to rely on Open Source Software (OSS) projects with many publicly available cases, problems, and code to investigate. They offer complex challenges going beyond typical local, university-driven projects. OSS projects are distributed and decentralised, utilising virtual teams in which participants can range from individual volunteers to professional development teams employed by a company. Furthermore, the practical relevance of OSS software is unquestioned, since OSS projects set de facto standards for software development in certain application domains, e.g., operating systems (Linux), web servers (LAMP stack), and mobile ecosystems (Android). Participating in OSS can benefit industry by leveraging development capacity for projects exceeding their own capabilities. In many cases, they must participate in OSS and build products on OSS platforms in order to access customers who are already using them. Learning how to function in this context, e.g., working in a self-organising virtual team requires particular knowledge and skills.

Therefore, OSS projects are fruitful grounds to set up a sophisticated teaching environment. Individual students could directly participate in a single project and investigate a specific problem, or, in order to achieve more demanding learning goals, groups of students participate through a collaborative program (*Richardson, Milewski & Mullick, 2006*; *Keenan, Steele & Jia, 2010*). For instance, students from several universities can participate in a common project pool (e.g., *Fagerholm et al., 2014b*; *Fagerholm, Oza & Münch, 2013*). From the industry perspective, OSS projects offer increased visibility and opportunities for recruitment, contribution to added features, and sustainability of key OSS components. From the teaching perspective, OSS projects provide a realistic learning experience with large software systems, and allow experiencing many aspects of collaborative software development. For researchers, OSS projects have a large amount of data available, with easier access than from companies' internal projects.

Data from OSS projects has been used for several purposes including improvement of teaching. Other data sources are also available for analysis, providing evidence-based means to improve teaching. An emerging trend is using learning analytics for supporting good

learning outcomes, to better understand learning progress, and to construct student profiles for tailored teaching. Ideally, this can allow real-time reaction to improve learning outcomes and to allow larger masses of students to participate in courses with limited teaching resources. The main benefits are, for instance, to address the drop-out problem and to provide more customised teaching for individuals. At the University of Helsinki, an example of a platform allowing learning analytics is the mooc.fi online learning environment, which provides courses on cyber security, programming in several languages, web service development, and algorithms. Research on the platform has, for instance, contributed methods to identify students based on typing patterns (*Longi et al., 2015*), which can help prevent cheating in an online environment, and to identify students in need of assistance, which allows increasing guidance for struggling students early on, and providing more challenging assignments for high-performing students (*Ahadi et al., 2015*).

# A GUIDELINE FOR INTEGRATING EMPIRICAL STUDIES WITH SOFTWARE ENGINEERING COURSES

In this section, we develop an experience-based guideline to integrate empirical studies with software engineering courses. We base the guideline on experiences gathered from our own software engineering courses, categorised from several perspectives. We first generalise common purposes, challenges, and validity considerations. These serve to determine the appropriateness of a particular study in a given context. We discuss appropriateness from two perspectives: (1) teaching at universities and (2) industry training. Finally, we share our experiences, discussing several aspects to be considered when integrating empirical studies with software engineering courses, e.g., motivation, scheduling, and effort.

## Purposes, challenges, and validity

To summarise the aforementioned kinds of empirical studies, we create the taxonomy presented in Tables 2–4. In this initial taxonomy, we include different purposes, challenges, and validity constraints to support the categorisation of study types, and the analysis of appropriateness in certain contexts. We identified a total of ten purposes, describing major learning goals associated with empirical studies in software engineering teaching that we consider important (Table 2). Complementing the purposes, we identified eight challenges that should be taken into account when designing empirical studies for educational purposes (Table 3). Purposes and challenges are intended to help teachers determine what study type is appropriate in a certain setting, e.g., does the actual setting allow for a (full) experiment, and if so, what challenges need to be addressed?

Apart from purposes and challenges, the quality of the outcomes of empirical studies—especially in the context of teaching using students as subjects (*Runeson, 2003*)—must be considered carefully. Taking the close relation to industry and relevance of the topics into account, we analysed the different study types for validity constraints. For example, researchers seek validity to solidify findings and to pave the way for generalisable knowledge, while industry is interested in business value. Furthermore, from a teaching perspective, result validity may be considered less important than achieving the learning goals. Therefore, we derived four validity considerations associated with empirical studies

**Table 2** Summary of learning goals and purposes for empirical studies in education.

| Purpose | Description |
|---|---|
| P01 | *Learn to formulate a research problem.* Students face a (real-world) problem that needs investigation. Therefore, the learning goal is to:<br>• Capture the problem.<br>• Formulate research questions.<br>• Formulate hypotheses regarding users or customers and their behaviour.<br>Due to the complexity of realistic, real-world settings, this task is demanding for, e.g., formulating a problem in a scientifically sound way, but keeping in mind the (industry) partners' needs. |
| P02 | *Learn to collect relevant data.* Collecting data in realistic settings is a demanding task, as data is usually scattered across different sources. The learning goal is to develop a meaningful data collection strategy that includes data from multiple sources within a setting and, optionally, backed up by further external data (from outside a given setting). |
| P03 | *Learn to analyse real-life data.* Real world situations are often incomplete or confidential thus hampering data analyses. The learning goal is to develop a data analysis strategy to overcome limited data. |
| P04 | *Learn to draw conclusions.* Based on collected and analysed data, the overall learning goal is to draw conclusions. Thus, in the (realistic) setting, students need to learn to:<br>• Gather empirical evidence on which conclusions are based.<br>• Test theories and/or conventional wisdom based on evidence.<br>• Draw conclusions from (limited) data and develop a strategy to utilise findings in practice.<br>The purpose is to gather findings or evidence, and to analyse the findings for relevance in the respective setting. Eventually, findings must contribute to solving the original problem and, thus, another learning goal is to develop transfer strategies to support utilisation of the findings in practice. |
| P05 | *Learn to experience and solve a specific problem.* A major purpose is to cause people to experience certain situations and to develop situation-specific solution strategies or approaches. This leads to:<br>• Experience regarding the problem-solution relation, e.g., understanding of the relationship between user behaviour and software design.<br>• Increased knowledge about a problem (domain).<br>• Increased knowledge about technology and methods.<br>• Increased knowledge about potential/feasible solutions and/or solution patterns.<br>Skills addressed by this learning goal are basic prerequisites that allow for developing solutions in general, as these skills address a specific problem, but also allow for developing transferable knowledge that can be applied to different contexts. |
| P06 | *Develop a software artefact.* In software engineering, software artefacts, especially prototypes, serve the (early) analysis of a specific problem. For this, prototypes allow for implementing and demonstrating solution strategies. The learning goal thus comprises:<br>• Create a (software) prototype to demonstrate a solution approach/strategy (feasibility study).<br>• Create artefacts to elaborate potential solution approaches/strategies for dis-/advantages (comparative study).<br>• Create artefacts to establish (quick) communication and feedback loops.<br>Software artefacts in general and prototypes in particular serve the elaboration of a problem, and help to understand the potential solutions. That is, such artefacts pave the way to the final solution. |
| P07 | *Coaching.* Another learning goal is to make stakeholders familiar with new methods and tools. Hence, utilisation of the new methods/tools need to be trained, i.e., develop and train necessary skills. |
| P08 | *Change of culture.* Continuous experimentation comprises a number of the other learning goals. However, continuous experimentation is more of a general organisational question than a project-specific endeavour. Therefore, utilising continuous experimentation also implies a cultural change toward experimentation in the implementing organisation. |
| P09 | *Learn about the impact.* Specific behaviour or decisions impact a system and/or a team, e.g., changing requirements or fluctuation in team composition. Therefore, it is important to learn about the effects that certain behaviour and decisions have in large and/or dynamic contexts. |
| P10 | *Learn about long-term effects.* Apparently "local" decisions might cause "global" effects. Thus, it is important to know about the long-term and/or snowballing effects caused by single decisions, e.g., a shortcut in the architecture leads to increased maintenance cost (technical debt). |

**Table 3** Summary of challenges that empirical studies in education face.

| Challenge | Description |
|---|---|
| C01 | *Finding or creating relevant cases.* The major challenge is to find and define proper and relevant cases, which bears some risks:<br>• A case may become irrelevant while conducting a study (e.g., changing environment, changing context parameters).<br>• A study might go into an unexpected direction (learning curve and, in response, focus shift).<br>• A relevant case must be narrowed down to the participating subjects, e.g., students have different skills and goals than professionals.<br>Cases must be balanced, e.g., learning goals must be achieved regardless whether the original case loses its relevance (procedural over technical knowledge) or students need to finish a thesis regardless of whether industry partners can apply study findings. |
| C02 | *No guaranteed outcome.* If a problem was found, there is no guarantee that a study will lead to an outcome. Furthermore, immediate applicability of the outcomes is not guaranteed, which means extra work for industry to transfer results into product development. |
| C03 | *Time constraints.* Apart from the appropriateness of the actual problem, time constrains limit the study. Time constraints can occur as:<br>• Limitations dictated by the curriculum/course schedule.<br>• Limitations dictated by industry schedules, e.g., product development cycles.<br>• Limitations dictated by individual schedules, e.g., students that are about to finish their studies.<br>Therefore, time constrains, together with resource limitations, define the basic parameters that affect the study objects (problem, potential/achievable solutions, completeness of results, validity of outcomes, and so forth). |
| C04 | *Resource limitations.* Studies require resources and, thus, availability of resources limits the study. Resource limitation can occur as:<br>• Availability of (the right) students, e.g., if a study requires students with a specific skill profile.<br>• Motivation of students to participate in a study (personal vs. study goals).<br>• Availability of industry resources (personnel tied to a study).<br>• Options to adequately integrate the study with (running) company processes.<br>Especially the availability is a critical factor. For instance, while one experiment consumes resources once, repetition and replication require a long-term commitment regarding resource availability, which implies significant investments of time and/or money. In order to make resources available, participating partners need to receive a sufficient benefit, which is often hard to define in empirical studies. |
| C05 | *Limited access to data.* Although it is one purpose in terms of learning goals, defining adequate hypotheses and variables that can be investigated in a course is challenging. Proper measurements must be defined, taking into account that potentially not all data is available, e.g., confidential data. Especially access to user data is challenging (a way out could be utilising OSS projects), as this data is usually strictly confidential. |
| C06 | *Built-in bias.* A special problem is bias. Each particular setting comes with an inherent set of biases, e.g.:<br>• Students' special skills affect the study, and students that are trained in advance of the study affect the outcomes.<br>• Too much or too little context knowledge of the subjects affects the study.<br>• Competing goals of the participants (especially students vs. practitioners) affect the study, e.g., students might try to optimise a study to achieve better grades while compromising the study goals.<br>Empirical studies suffer from certain limitations, and in the context of teaching, special attention needs to be paid to bias and threats to validity. |
| C07 | *Communication.* Empirical investigations create knowledge, data, and potentially software artefacts. Therefore, results need to be quickly communicated to the participants. Quick feedback helps to, e.g., determine the relevance of results, appropriateness of the instrument, and determining necessary adjustments. Thus, fast feedback loops are necessary. |
| C08 | *Creating a Simulation Model.* For simulation-based research/teaching, setting up a simulation model is a demanding task, which consumes time and resources thus generating cost. The entire domain under consideration must be captured to create a model that allows for generating useful data. |

**Table 4** Summary of validity considerations when using empirical studies in education.

| Validity consideration | Description |
|---|---|
| V01 | *Emphasis: meeting teaching goals.* Use is valid if procedural learning goals are met. Validity of conclusions and usefulness for industry are of secondary importance from the teaching perspective. |
| V02 | *Emphasis: meeting business or organisational goals.* Validity depends on what value is created for the business. Direct business value is rare; a more likely result is increased knowledge of the problem area, technology, work methods, or potential solutions or solution patterns. |
| V03 | *Emphasis: creating a sound study design.* Focus is on the internal validity, and results of a study are "side effects". If the learning goal is to understand experimentation itself, internal and external validity could have higher relevance. |
| V04 | *Emphasis: meeting research goals.* Especially in simulation, the validity of the gathered data depends on the quality of the simulation model, and also on the quality of the simulation environment. |

for teaching from the desired learning outcomes and our knowledge about the educational context (Table 4).

## Establishing context and goals, and determining a study type

We provide an initial assignment to the four major study types *experiment*, *case study*, *simulation*, and *continuous experimentation*. Individual studies are left out, since the particular challenges result from the concrete instrument applied in the respective study, e.g., an individual study may implement an experiment or a case study.

Table 5 provides an initial assignment of purposes, challenges, and validity constraints from the academic perspective, while Table 6 provides the industry perspective. The tables support decision-making when selecting appropriate instruments. For instance, if the context is university, and students shall learn to solve a particular problem (P05) by developing a software tool (P06), teachers should opt for a case study. In an industry context, both case studies and experiments can be utilized. However, as Table 6 illustrates, an industry experiment is more demanding, with more challenges to address, e.g., built-in bias (C06) and communication (C07), and a different validity emphasis (V02). Another observation is that there are no differences suggested between the two settings in the case of the simulation instrument. In both, the major challenge is the simulation model, which affects learning, effort for its creation (C08), and validity constraints (V04).

Concrete goals must be considered and balanced alongside contextual information when selecting a particular study type. This also includes goals going beyond classic learning goals. For instance, all stakeholders (students, teachers, and industry partners) come into contact when performing case studies in industry, allowing for several opportunities. Students make contact with industry and could find a job. Students team up with other students to create an idea which may eventually lead to the founding of a company. On the other hand, industry can conduct research cheaply, as they usually pay with time spent, sponsor an idea, or pay a small fee to keep a software engineering lab running. In either case, industry gets access to the latest knowledge and fresh resources. Finally, researchers have the opportunity to conduct some research (given the limitations mentioned above).

**Table 5  Education in academia/university.**

| | Experiment | Case study | Continuous experimentation | Simulation |
|---|---|---|---|---|
| Purpose | P01, P03, P04 | P01, P02, P03, P04, P05, P06 | P01, P02, P03, P04 | P09, P10 |
| Challenges | C01, C03, C04, C05 | C01, C02, C03, C04, P05 | C01, C03, C04, C05 | C01, C05, C08 |
| Validity | V01 | V01, V02 | V03 | V04 |

**Table 6  Education in industry.**

| | Experiment | Case study | Continuous experimentation | Simulation |
|---|---|---|---|---|
| Purpose | P04, P05, P06 | P05, P06 | P02, P03, P06, P07, P08 | P09, P10 |
| Challenges | C01, C03, C04, C05, C06, C07 | C01, C02, C03, C04 | C03, C04 C05, C07 | C01, C05, C08 |
| Validity | V02 | V02 | V02 | V04 |

## Motivating students to conduct empirical studies

Making contact with industry and the prospects of finding a job or starting a company, foster students' motivation to actively participate in courses, and may contribute to higher motivation, engagement, and better understanding of course contents. For instance, in *Kuhrmann, Fernández & Münch (2013)*, we reported on a new teaching format applied to a software process modelling course, including an empirical study. The course evaluation showed that students experienced the course significantly better than the class before (without the study), although they perceived the course as more demanding (see 'Example 1: a course on software process modelling with and without experiments'). The evaluation showed that students understood contents and their relevance better, gathered advanced knowledge and learned to apply it experiencing practical effects, e.g., consequences of wrong design decisions. Our experience also shows that encouraging students to develop ideas and create products boosts motivation (c.f. *Brügge, Krusche & Alperowitz, 2015*). For instance, smart phone apps can be developed in collaboration with industry partners and published in an app-store. Gaining visibility, real clients, real feedback, and real bug reports guides students through the whole software development and product life cycle.

Nevertheless, apart from all potentially positive motivating drivers, a major driver for students is to get the best possible grade. Also, the number of credits must reflect the effort required to conduct the study. For students, the amount of time required to receive a credit point is an important consideration. Since empirical studies are demanding in terms of effort, and credits form the compensation, software engineering courses that include empirical studies must adequately "remunerate" the students for their efforts.

## Scheduling

Having defined the goals and acquired (motivated) students and, optionally, partners from industry, the challenges C03 and C04 (Table 3) must be addressed. Planning empirical studies in a standard university curriculum is demanding, as students usually take several courses thus having limited resources. Furthermore, courses often span 12–15 weeks and if industry is involved, their schedules must be respected as well. In *Kuhrmann (2012)*, we

provided a generic template that integrates classic teaching with explicit workshop slots, which can be used to conduct empirical studies. In *Kuhrmann, Fernández & Münch (2013)* and *Kuhrmann, Femmer & Eckhardt (2014)*, we provided concrete instances and reported on the feasibility of the proposed template. However, conducting empirical studies in collaboration with industry requires refining the generic template. We consider three basic planning patterns appropriate:

- *Workshop model:* In the workshop model, teachers, students, and practitioners conduct a workshop in which they collaboratively work on a problem. An example is a lab-based environment, such as the Software Factory (*Fagerholm, Oza & Münch, 2013*). Moreover, the workshop model is quite common in industry training (usually 1–5 days); a study that fits that schedule is more likely accepted by industry partners.
- *Interleaved model:* The interleaved model allows conducting a "long-running" study. Normal work slots alternate with workshop slots, e.g., a new method is deployed and trained, practitioners apply it, researchers evaluate, improve and/or train new aspects of it, practitioners continue application, and so forth. Furthermore, this model proved beneficial when supervising students conducting individual studies in industry. There are several benefits: regular work is not disturbed over a long period; training can be done iteratively; and cases can be observed over a longer time period. However, course schedules limit the applicability with student groups.
- *Observation model:* This model is the classic research model adopted for education purposes. Students or practitioners are instructed, work independently on a task, and get coaching from teachers. Besides the coaching, teachers monitor the correct application of empirical methods to collect and analyse data.

Planning the study and justifying the study plan with all time constraints needs to be done carefully, and requires the commitment of all participants to ensure availability of personnel and resources.

## Further study type selection criteria

Apart from the criteria already discussed, we wish to highlight some further criteria that may influence the selection of study types for educational purposes. First, in Table 7, we summarize well-known criteria from literature (e.g., *Wohlin et al., 2012*) and further criteria that we consider relevant for study type selection. The table includes an experience-based rating for the criteria. However, this rating has to be considered as a subjective recommendation, as it is hard to precisely define, e.g., the degree of motivation or student satisfaction.

We note that the knowledge and skill level of students should also be taken into account when selecting and tailoring an empirical instrument for teaching. In Table 8, we provide an experience-based assessment of how different study types can be adjusted for different levels of students. Two student levels are considered: Bachelor's (0–3 years of study) and Master's (3–5 years of study) levels. In industry, these may be interpreted either based on employees' level of education or their working experience in the field. The primary means of adjustment is selection of a suitable problem scope and setting expectations for

Fagerholm et al. (2017), *PeerJ Comput. Sci.*, DOI 10.7717/peerj-cs.131

**Table 7  Further study selection criteria for different study types. Each study type is ranked relative to the others on three levels and may span more than one level (LO: low, ME: medium, HI: high).**

| | Experiment | | | Case study | | | Continuous experimentation | | | Simulation | | | Individual studies | | |
|---|---|---|---|---|---|---|---|---|---|---|---|---|---|---|---|
| | LO | ME | HI | LO | ME | HI | LO | ME | HI | LO | ME | HI | LO | ME | HI |
| Degree of execution control | | | × | × | × | | | | × | | | × | × | × | × |
| Degree of measurement control | | | × | | | × | × | × | | | | × | × | × | × |
| Degree of validity | | × | × | × | × | × | | × | × | × | × | | | × | |
| Motivation to participate in a study | × | × | | | × | × | | × | | × | × | | | | × |
| Motivation created by the study | | × | × | | × | × | | | × | × | × | | | × | × |
| Student satisfaction | | × | × | | × | × | | | × | × | × | | | × | × |
| Scheduling effort | | | × | × | | | | | × | × | | | × | | |
| Ease of goal definition | × | | | | × | × | × | × | | | × | × | × | × | |
| Effort to prepare/conduct a study | | | × | × | | | | | × | × | | | × | | |

**Table 8  Adjusting study types to student levels. Length of study is indicative and based here on European standards.**

|  | Bachelor's level (first 3 years of study) | Master's level (between 3–5 years of study) |
| --- | --- | --- |
| Experiment | Simple experiments with few variables. Experiment design given. | More complex multivariate experiments. Own experiment design. |
| Case Study | Limited topics, restricted to chosen context, few informants. Little or no generalisation. Exploratory, descriptive or intrinsic case studies. | Topics related to well specified software engineering areas. Some generalisation. Limitations of generalisation fully analysed. All case study types. |
| Continuous Experimentation | Rudimentary practice with synthetic scenarios. Focus on understanding basic steps such as identifying assumptions, creating hypotheses, and collecting data. | More advanced scenarios or limited real-life experiments. Focus on drawing conclusions from data and understanding limitations. |
| Simulation | Using ready-made simulation models and given data to explore topics through simulation. | Exploring the effect of changes in models using given data or how ready-made models behave with student-collected data. Some exploration with creating simulation models. |
| Individual Studies | Focus on finding and summarising existing research. | Focus on answering specific research problems by applying existing research and own data collection. No requirement of scientifically novel results. |

appropriate result scope. For example, case studies at the Bachelor's level can be more limited in scope and focus on exploratory, descriptive, or intrinsic designs without much generalisation beyond the case environments. On the Master's level, some generalisation can be expected although still limited. Assessment of the possibilities to generalise can be expected at this level. This assessment must be considered as a subjective starting point for adjustment, as students are different and educators should, as far as possible, tailor courses for individuals in order to provide the best opportunities for learning.

## EXPERIENCES

In this section, we provide some experiences gathered from implementing empirical instruments in university teaching. We provide selected examples, outline the respective courses (purpose, approach, outcomes), and provide feedback and evaluation (formal as well as informal) to reflect the students' perception of these courses.

### Example 1: a course on software process modelling with and without experiments

A course on software process modelling, which implements the curriculum presented in *Kuhrmann, Fernández & Münch (2013)*, serves as first example. The course was offered multiple times at the Technische Universität München (TUM) and the University of Helsinki. In Munich, after the initial run, the course was reorganized according to the concept presented in *Kuhrmann (2012)* in which we presented an approach to integrate experimentation with practical Software Engineering courses. Due to the reorganization, students experienced the (abstract) topics while conducting a controlled experiment on which we reported in *Kuhrmann, Fernandez & Knapp (2013)*. Moreover, due to the

**Table 9** Formal evaluation (anonymous questionnaire, comparison winter 2010/2011 and 2011/2012, TUM, result interpretation: ↑ large improvement, ↗, small improvement; →, no change; ↘, small deterioration; ↓, large deterioration.)

| Criterion | Winter 2010/2011 | Winter2011/2012 | Result |
|---|---|---|---|
| Number of completed questionnaires | 6 (9 participants) | 8 (14 participants) | – |
| *Common criteria (1 = very high, 5 = very low)* | | | |
| Complexity | 3.00 (old questionnaire: "level") | 2.38 | −0.62 ↑ |
| Volume | 2.83 (old: one question) | 2.12 | −0.71 ↑ |
| Speed | | 2.75 | −0.08 → |
| Appropriateness of effort compared to ECTS points | n.a. | 3.00 | n.a. |
| *Overall rating (1 = very good, 5 = very bad)* | | | |
| Lecture | 1.25 | 1.5 | +0.25 ↘ |
| Exercise | 2.17 | 1.33 | −0.84 ↑ |
| Relation to practice | 2.0 | 1.62 | −0.38 ↗ |

repeated execution in which we applied a course structure both without *and* with empirical instruments, we can present a number of experiences and a comparison.

### Formal evaluation

In Table 9, we present the comparison based on the formal course evaluations conducted by the Faculty of Informatics at TUM. Due to updated questionnaires, the evaluations are not directly comparable. However, the basic information can still be extracted. The formal evaluation shows a significant increase of the scores[1] regarding exercise quality and relation to practice, although, at the same time, the students also see the lecture as more demanding. Since the basic course contents did not change, we interpret this evaluation as an increased awareness toward the course topic, which might be caused by the stronger utilization of practical aspects through the experiment. We see this as an indication that introducing experiments could have a positive effect, although a full validation is missing.

### Informal evaluation

Besides the formal faculty-driven evaluation, we also performed two informal feedback rounds in the course instance in which we adopted the empirical instruments. We asked the students to write a *one-minute-paper* that contained the following three questions to be answered in short words:

1. (up to 5) points that are positive
2. (up to 5) points that are negative
3. (up to 5) points that I still wanted to say (informal)

Table 10 shows the summarized results from the informal evaluation: The structure of the class, the selected topics, the combination of theory and practice, and the way of continuously evaluating the work and finding the final grades were rated positively. Especially the practical projects and the team work in the workshops were highlighted. On the other hand, students mentioned the tough schedule and the not always optimal tailoring of tasks[2] for the practical sessions.

[1]Note that in Table 9, smaller scores are better.

[2]Since we informed the students about the "experimental" character of this special course in advance, the students did not complain, but welcomed the opportunity to give the feedback to improve their own class.

**Table 10 Summarized evaluation of the one-minute-papers (winter 2011/2012, TUM).**

**Positive Aspects**

Structure of the topics and the class,

Combination of theory and practice,

Projects in teams (atmosphere),

Self-motivation due to presentations,

Continuous evaluation and finding of the final grades

**Negative Aspects**

Tough schedule,

Tailoring of the tasks for the practical sessions was not always optimal

Students signed off, just because of the examination procedure

**Informal**

"Thank you, this was the lecture I learned the most."

"Super class, and I loved those many samples from practice."

## Example 2: a course on agile project management and software development with experiments

The second example is an advanced course on agile software project management, which is also grounded in the general course pattern presented in *Kuhrmann (2012)*. A detailed description of the course and data obtained from the experiments is provided in *Kuhrmann, Femmer & Eckhardt (2014)*. In this course, offered at the Technische Universität München, the main purpose of the experiment instrument was to create awareness—scientific results were not the objective. We implemented two experiments:

**Experiment 1 (Group Dynamics)** The first experiment aimed at demonstrating how groups of people collaborate in teams under stress (*Kuhrmann & Münch, 2016b*). Therefore, we introduced the Tuckman Model (*Tuckman, 1965*), which describes group formation processes, and designed a simple experiment in which the students had to sort sweets and to document the outcomes. During the different experiment runs, we put some pressure on the students, e.g., increasing task complexity, enforced turnover, and external disturbances. Although this experiment did not aim at finding new scientific revelations, we could confirm the Tuckman Model and show that group performance suffers from turnover.

**Experiment 2 (Distributed Development)** The second experiment was designed to give the students the opportunity to deal with hopeless situations (*Kuhrmann & Münch, 2016a*): we designed a Software Engineering "Kobayashi Maru" test.[3] Students were separated into two sets, each consisting of two groups for a total of four groups. Each group had to develop a very simple console-based chat application (requirements in the form of user stories and test cases were provided), the groups were separated (each was located in a separate room), and each group was allowed to use only one communication channel (e-mail and Skype respectively). After the task had been presented to them, the groups were immediately separated to avoid any direct communication, and for each group, a researcher monitored the compliance to the experiment rules. As the students

[3] In the Star Trek franchise, Kobayashi Maru is a leadership test with a no-win scenario; see https://en.wikipedia.org/wiki/Kobayashi_Maru.

**Table 11** Formal evaluation (anonymous questionnaire, winter 2012/2013, TUM).

| Criterion | Winter 2012/2013 |
|---|---|
| Number of completed questionnaires | 15 (19 participants) |
| *Common criteria (1 = very high, 5 = very low)* | |
| Complexity | 3.27 (just right) |
| Volume | 2.93 (just right) |
| Speed | 3.07 (just right) |
| Appropriateness of effort compared to ECTS points | 3.07 (just right) |
| *Overall rating (1 = very good, 5 = very bad)* | |
| Lecture | 1.33 |
| Exercise | 1.40 |
| Relation to practice | 1.57 |

**Table 12** Summarized evaluation of the one-minute-papers (winter 2012/2013, TUM).

| Informal |
|---|
| "Course discusses topics of interest from practical point of view." |
| "I like the practical approach of teaching." |
| "By far one of my favorite courses at all. Very interactive and relaxed atmosphere. Great exercises." |
| "Interactive, student presentations, experiments." |
| "Applicability of the course immediately in my work for other software projects." |

did not have the chance to initially find some agreements, the projects were failures-by-design. The students immediately started to work (they had only 90 min to develop the working software), yet, nobody came up with the idea to negotiate a communication protocol first. Therefore, after the deadline, no group could show any working software. In a closing feedback session, we revealed the nature of the experiment and discussed the observations.

### Formal evaluation

In Table 11, we present the comparison based on the formal course evaluations conducted by the Faculty of Informatics. Although we have only one evaluated instance of this course, we use the same structure as in Table 9 to present the data. The evaluation shows this course to be on approximately the same level as the improved software process modelling course.

### Informal evaluation

Besides the formal faculty-driven evaluation, we again performed two informal feedback rounds in the course. We asked the students to write a *one-minute-paper* (see above). Since the outcomes are actually the same as already presented in Table 10, we only present the informal comments (third question) in Table 12.

## Example 3: master's theses using a case study approach

Master's theses provide opportunities for individual students to apply a specific research approach to a chosen problem. The third example comes from a selection of Master's thesis projects which we have supervised at the University of Helsinki, all of which use a case study approach. They are therefore examples of both individual studies and case studies.

The first thesis project investigated a software prototype game and applied usability and user experience evaluation methods to determine whether it fulfilled two sets of criteria: the entertainment value of the game, and the ability to tag photos as a side effect of playing the game. The game itself was implemented by a student team in cooperation with a company, and the thesis writer was part of the implementation team. In this thesis, the game constituted the case and four sources of evidence were used: user interviews, in-game data collection, a questionnaire, and observations from a think-aloud playing session. The thesis can be characterised as an intrinsic case study (*Stake, 1995*; *Baxter & Jack, 2008*), since the objective was not to gain understanding of an abstract construct or general phenomenon, nor to build a theory. Rather, the case itself was of interest, and the results were suitable for making further decisions regarding development of a full game based on the prototype.

The second thesis project investigated continuous delivery and continuous experimentation in the B2B domain. The objective was to analyse challenges, benefits, and organisational aspects in a concrete company case. The thesis writer was an employee in the company and was thus a participant observer. In this thesis, the case consisted of the development process used by two teams for two separate software products. Two sources of evidence were used: participant observation and interviews with 12 team members, six in each of the two teams. The thesis can be characterised as an exploratory deductive case study, where the aim was to explore how continuous delivery and continuous experimentation could be applied in the company and what challenges and success factors are encountered. The thesis aimed to generalise and provide results that could be adapted to other B2B companies.

The third thesis project investigated the state of the practice of experiment-driven software product and service development. The objective was to understand the state of the practice of continuous experimentation and to identify key challenges and success factors in adopting continuous experimentation. The thesis can be characterised as a qualitative survey design, which resembles a case study but relies on a single source of evidence. In that sense, the thesis was close to an intrinsic case study, as it aimed to develop a multifaceted understanding of the topic rather than develop theory. The thesis used material from 13 interviews in 10 software companies. The result of the thesis was a rich picture of the state of the art concerning experiment-driven software development in the case companies. Although the primary aim was not to generalise, the results were relevant as comparison points in other companies.

### Informal evaluation

Utilising a case study approach in the Masters' theses provided the opportunity to investigate highly relevant problems in their natural context. Each thesis gained from

having an industrial connection which provided real-life constraints, questions, and data. In each thesis, the student had to consider the setting, objectives, questions, methods, data collection, and analysis procedures and adjust the general case study research method to their particular implementation. We observed high motivation among the students, timely completion of subtasks and of the thesis as a whole, and clear maturation with a complex individual project. Two of the theses were developed into scientific papers that have been published in peer-reviewed forums.

Based on these examples, the difficulties related to case studies can be summarised into three categories. First, finding and scoping a relevant research problem can be difficult for many students, as they do not have the necessary overview of the present literature that is needed. The role of the advisor is of prime importance in the beginning: helping to formulate the research questions and pinpoint what the case or unit of analysis is. Second, understanding case study research as a method can take a long time without proper guidance. Providing relevant method literature, identifying the key concepts, and providing an understanding of how to implement the method in practice —designing the study—are areas where the advisor can help. The data collection is usually interesting and straightforward, perhaps with some practical challenges related to finding data sources. As these can often be overcome by some persistence, the third category is related to performing the analysis and writing up the case report or thesis. Students do not often have a chance to practice these skills on a regular basis, and thus there are many questions regarding analysis choices and patterns for writing up results that an advisor may be able to help with.

Although we rely here only on informal evaluation, these examples have convinced us that case studies of different types are well suited as teaching tools. They require a wide range of skills which the students must acquire, and these skills are applicable in many other settings as well. Perhaps the most important insight to be gained from conducting case studies is that students are faced with a wide variety of data that challenge their preconceptions and develop their ability to observe phenomena in their real-life context.

## DISCUSSION

Implementing a course using empirical instruments provided us with a number of insights. Related to the scientific and organizational perspective, we learned that course preparation causes more effort compared to classic teaching. First, the examples and cases to be used in experiments need to be tailored accordingly: there must be time restrictions due to schedule requirements. This has two major impacts. First, the investigated topic is of reduced complexity, which causes it to be less realistic. Second, research questions must be carefully selected for reasonable treatment within time constraints. Therefore, we consider explorative (curiosity-driven) or confirmative experiments meaningful, i.e., experiments of low criticality.

From the teaching perspective, we experienced that the choice of a real world example rather than an artificial toy example has proved to be successful. For example, the experiment outcome from *Kuhrmann, Fernandez & Knapp (2013)* was a fully implemented process to which the process owner stated that he did not expect the student groups to create

"such a comprehensive solution in this little time." Another goal—"let students experience the consequences of their decisions"—was also achieved. For instance, in the course on software process modelling, while implementing the process in a workshop session, we could observe a certain learning curve. One team had a complete design, but selected an inappropriate modelling concept. Later, the team had to refactor the implementation, which was an annoying and time-consuming task, both increasing their awareness of the consequences of certain design decisions. Furthermore, students also experienced how difficult it is to transform informal information or tacit knowledge into process models. The students could also see how difficult it is for individuals to formulate their behaviour in a rule-oriented manner.

For the course on software process modelling in Munich, we compared the final grades of both courses and recognized significantly better grades in the second run. During course exams, the students could not only answer all (theoretical) knowledge-related questions, but also all knowledge-transfer and application-related questions. The students usually referred to the practical examples and were able to transfer and apply their experiences to new situations.

Finally, the case study-based Master's theses allowed our students to be embedded in projects with real-life connections. Apart from their educational value for the students, they contributed to the scientific literature and helped students in their early careers. Although our industry connections were important in obtaining the cases, the students themselves learned to be self-directed in their work and gained significant domain knowledge. As thesis supervisors, we found that there was some additional effort in introducing case study methodology to students—methodology courses do not fully prepare students to actually carry out a study of their own, which is to be expected. However, being embedded in the project and receiving feedback from the project environment and its stakeholders meant that it was easy to convince students of the necessity of a structured approach. Once students were up to speed, the extra supervision effort was compensated by more autonomous work on the students' part.

## Limitations

The guideline presented in this paper has not been systematically tested in different learning environments. Instead, it represents a starting point based on reflection grounded in teaching practice. We consider the limitations of the study in terms of qualitative criteria for validity (c.f. *Creswell, 2009*).

Internal validity concerns the congruence between findings and reality. In this study, internal validity then concerns how credible the guidelines are in light of the realities of software engineering education. As that reality is constantly changing, the match between guidelines and teaching can never be perfect. Our study has applied triangulation to increase the internal validity of the results. We have utilised several types of teaching in different modes and in different universities, and with different teachers, to obtain a richer set of experiences to draw guidelines from.

External validity refers to the extent to which findings can be applied to other situations. As our aim is not theory testing, external validity in this article is about enhancing, as far

as possible, the transferability of the results. We argue that the guideline developed herein covers a wide range of teaching and learning situations, and thus can be applied widely in graduate and undergraduate education in software engineering. We have attempted to elucidate the limitations of applying the guideline by mapping study types differently to education in academia and industry, and to different purposes, challenges, and validity concerns of interest to teachers. In addition to these limitations, we see that there are certain situations where the guideline would be unsuitable. First, when the execution of an empirical study would cause ethical problems or legal consequences for any of the involved parties; in this case, the teacher should direct the student to a different task. Second, the guideline relies on the teacher to assess whether a particular student possesses the necessary prerequisite skills to carry out a particular study; the guideline is not transferable if that information is missing. Third, the guideline makes certain assumptions about the learning environment, such as availability of industry partners for Master's degree projects, and the availability of certain teaching resources for other study types. When attempting to apply the guideline, teachers should consider whether the necessary resources are available.

## CONCLUSION

There is a lack of guidance on how to use empirical studies in software engineering education. In order to address this gap, this paper provides an overview of different types of empirical studies, their suitability for use in education, as well as challenges with respect to their execution. We analysed our own teaching and the different studies that we applied as part of it, and reported on selected studies from existing literature. Rather than having students conduct pure research, we opt for including different empirical instruments into software engineering courses as means to stimulate learning.

The present paper provides an initial systematisation of empirical instruments from the educational perspective. We derived a set of purposes and challenges relevant for selecting a particular study type. Furthermore, we also discussed validity constraints regarding the results of course-integrated studies. Based on our experiences, we assigned the different purposes, challenges, and validity constraints to the different study types, and we provided further discussion on motivation and scheduling issues. We also defined a set of further study selection criteria to provide an initial guideline that helps teachers to select and include empirical studies in their courses. We believe the guideline could be used in a wide variety of settings. We note that the guideline is limited in that it considers a limited number of study types and learning outcomes—those that they authors have experience with as teaching aids and study purposes. They may not be suitable in situations where significantly different study types or learning outcomes are called for. Since, to the best of our knowledge, no comparable guidelines exist, we cordially invite teachers and researchers to discuss and improve on this proposal. In particular, future work could focus on applying the guidelines in different kinds of software engineering courses and programs, both within academic university education and in industry training. The purposes, challenges, and constraints presented here could thus be further validated, refined, and perhaps extended.

Another particular consideration is how to perform student assessment when using empirical studies for educational purposes, in particularly when group work is involved.

What should be assessed, how should assessment be performed fairly when many students are involved, and how should, e.g., knowledge of empirical methods, domain knowledge, procedural knowledge, and the quality of outcomes be balanced in the assessment? We believe that the purposes and validity considerations in Tables 2 and 4 could serve as a starting point for creating rubrics that are relevant for this type of teaching.

Finally, further studies are needed to test the effectiveness of courses using the proposed approaches in terms of their ability to teach. The learning outcomes of such courses should be further explored: beyond what is currently known, what do students learn by conducting empirical studies, and how do their learning outcomes differ from other approaches to software engineering education?

### Funding
This work was supported by Tekes, the Finnish Funding Agency for Technology and Innovation, as part of the N4S Program of DIGILE (Finnish Strategic Centre for Science, Technology and Innovation in the field of ICT and digital business). The funders had no role in study design, data collection and analysis, decision to publish, or preparation of the manuscript.

### Grant Disclosures
The following grant information was disclosed by the authors:
Tekes, the Finnish Funding Agency for Technology and Innovation.

### Competing Interests
The authors declare there are no competing interests.

### Author Contributions
- Fabian Fagerholm, Marco Kuhrmann and Jürgen Münch conceived and designed the experiments, performed the experiments, analyzed the data, contributed reagents/materials/analysis tools, wrote the paper, prepared figures and/or tables, performed the computation work, reviewed drafts of the paper.

### Data Availability
The raw data is included in the tables.

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
