# Peer review of "Guidelines for using empirical studies in software engineering education"

_PeerJ Computer Science, doi:10.7717/peerj-cs.131_

## Round 0.1 · original submission · Major Revisions

· Academic Editor

Major Revisions

All three reviewers are positive about the paper, so I very much hope that you will be able to engage with their constructive reviews and produce a revised version quite soon. Two of them marked "major revisions", however, feeling that their comments really need to be addressed and checked before publication, so I have gone with the same decision. It seems to me that the main work needed is to clarify the scope and contribution of the paper, picking a better title to reflect it, ensuring that there is enough detail (especially in Section 5) to enable readers to understand what you have done, and focusing the related work section appropriately to position your contribution and also point readers usefully to other work they might find useful. At the same time, you should avoid over-claiming on the basis of a small study, and distinguish conclusions that are suggested and which you think will prove to be true from those for which there is conclusive evidence. Further points will be found in the reviews. I note that they do not concur on every point, but I think you will be able to make use of them.

Reviewer 1 ·

Basic reporting

The English is generally good. The structure is less so and the balance between the trivial and the interesting should be changed.

Overall it's ok but there's plenty of scope for improvement.

I give more detailed suggestions in the General Comments

Experimental design

The paper brings together some relatively novel ideas and is / could be a stimulating read.

The main approach is advocacy and author experience. More objective evidence might be added but even if it wasn't the paper would generate discussion and so could be published.

Validity of the findings

The worth of the paper stands or falls by the ideas. There are few statistics.

Comments for the author

The title should be changed as the paper describes many things in addition to experiments e.g. simulation and case studies.

Overall the paper brings together some relatively novel ideas and is thought provoking so I believe it should be published but there are many opportunities for improvement.

(i) There’s a lack of structure, other than around different forms of empirical research. I think the authors struggle to shed their perspective of empirical software engineering researchers. This paper is about pedagogy. And what about different forms of learning, different types and levels of student. In any case is this always the answer, sometimes the answer? Indeed answer to exactly what question? (Potentially Tables 5 and 6 might serve that purpose but it needs moving to near the start of the paper and I’d think about some additional dimensions such as type of student/course etc.

(ii) I would prune the generic material and try to focus on teaching examples. It would be very interesting to know how some of these ideas were implemented. What went well and what less so? Presently the reader has to wait until Section 5 and that's extremely brief.

(iii) I think the claims for the benefits of this type of teaching need to be sharpened. "In our experience, this approach prepared students 36 better for later working life” l35-6. So better than what? How do you know? How might this be demonstrated (perhaps in the future)? Likewise claims of causality "Training in empirical instruments, such as experimentation ... cause students to become more …”. Such claims could (should?) be put forward but you need to be clear that you’re essentially using rhetoric or advocacy. Again right at the end of the paper are a few references to improved results (ll596-600) but it’s all too late and too brief.

However I do agree that ”little attention has been paid to using empirical studies with an educational purpose in mind.” This is an opportunity and the paper is useful in provoking thought and discussion around this topic.

DETAILED REMARKS:
The Related Work section could be improved. It seems to wander around and then keep repeating that the literature thus far really only considers students as participants research experiments. It might help to try to think about types of education and learning goal.

Also a good deal of material is a rather generic discussion of different empirical research methods; what would be more interesting is a detailed evaluation of how such methods can be used for teaching purposes and also some examples. There are some tantalising references to examples such as continuous experimentation with student groups (ll264-9) but the reader learns almost nothing without going to the original article. Even by going back to the original Kuhrmann et al. (2014), article I didn’t get a very clear idea of how the treatments were identified, students allocated, how the debriefing work and on what basis this was deemed a pedagogic success. (I’m sure it was very engaging and quite likely valuable but it would be nice to have some reasons to support this expectation.)

If simulation is used for teaching how do you ensure the model is sound / valid? What situations are best / least suited for simulation. I note that later on, Table 3 (C08) also comments on this problem. In what sense is this experimentation?

It’s unclear how individual studies are a form of experiment. Please explain more clearly what is meant here as I can’t really see how they differ from a standard piece of coursework.

How does the discussion on replication impinge on teaching?

What are the Pn’s in Table 2? Teaching objectives? Also there seem to be a lot of overlaps between the purposes. Part of P01 is capturing the problem. Doesn’t this entail collecting relevant data? What’s the difference between impact and changing the culture. Perhaps there are some missing higher level concepts? There are also some time and other dependencies.

Why are Tables 3 and 4 distinct, i.e. what’s the difference between a challenge and validity consideration?

The coclusions should also consider when not to use these ideas.

TYPOS:
l63: "An example of a disciplines that already has a high level”?

Table 3: C06 " Bias is automatically introduced by the respective settings,” ???

l632 Incomplete reference

·

Basic reporting

The authors wrote a well structured paper on their approach introducing empirical research methods into their software engineering (SE) courses. They were able to identify purposes, challenges and validity considerations for the use of empirical studies in SE education. They suggest a mapping of these identified matters to the four major research approaches : experiment, case study, continuous experimentation and simulation. They motivate their findings based on the course evaluation data from their students. The authors conclude their students are more motivated and get higher scores when using empirical studies in education.

Title: I think the title is misleading. I suggest the authors change it to one that fits better to the aim of the paper: ‘The present paper provides an initial systematisation of empirical instruments from the educational  perspective’. The paper does not only provides a experiment-driven approach and does not really go into detail about how to setup a course to use this experiment driven approach as the current title could suggest.

Introduction: misplaced comma (line 16). Line 29-31: seems somewhat in contradiction to say it ‘beats empirical evidence’, but that same evidence is not available. Line 52-55 mention the sections, but leave out section 5. Section 6 is mentioned, but says ‘Conclusion’. The section in the paper is named ‘Summary’. I suggest the authors change it to ‘Conclusion’. 

Related work: The related work is mostly linked to motivation of students. Did the authors search for any literature on the long term effects that they mention in the introduction(line 17/18)? Why did the authors not mention anything on educational approaches such as evidence based learning, problem based learning or project based learning? The last one for example is used a lot in Europe (e.g. The Netherlands, Sweden, Belgium, Germany, France …). During the projects often case studies are used that come from industry. In these projects different research activities are trained (for example: analysis). 

Section 3 - Overview: the overview is clearly written. The examples that are given are nice and informative. How did the authors came to this set of types? This does not cover al empirical study types. Did they consider other types? Line 260-261: ‘From the teaching perspective, continuous experimentation helps understand the connection between  software development techniques and business.’ Do the authors have any experiment-data / argument / reference to support this?

Section 4 - Guideline: Line 425 - 426 ‘were extracted .. ‘ which studies were used? Home many?
It is not clear how the tables 2,3,4 were constructed. Why are these categories / criteria used? Was there discussion between several experts and/ or industry? In tables 5 and 6 a mapping is presented. How was this mapping done? What where the criteria? For example, I can imagine that P01 and P03 also can be learned during a case study. Or do I mis some information here?
4.3 ‘The factors’ : which factors?
The authors could have elaborated more on the process of setting up a course that uses empirical studies. The section focus mostly on which study type to choose. Can the authors provide a step by step approach for the educator?

Sections 5 - Experience : Line 559 - 560. The description of the experiment is not completely clear. The students used communication, but communication was avoided? (The experiment sound great by the way). Line 585 / 586: not surprising. Line 587-595: This part is a little to vague: ‘certain learning curve’, ‘inappropriate modelling concept’, certain ‘certain design decisions’ . What were the key matters that lead to the students’ awareness?

Summary: ‘Using empirical studies in SE education is not yet common ..’ I disagree. As mentioned above there are educational approaches that already use components of empirical studies sinds years. I do agree it is not often written down and indeed it lacks of guidance for educators in SE. In my opinion the conclusion that the authors’ experience ‘shows’ increased motivation and participation and better quality is too firm. The number of participants is to low to really state that and generalise it. On the other hand the initial framework / guideline that they present seems to be a very valuable contribution.

References: Line 632: Missing title

Experimental design

One one hand this paper does not really provides an experiment. On the other hand the authors do analyse participants of several courses where a certain method was used and use the results to argue the use of empirical studies improves different matters. In that case the authors could have elaborate more on the background of the participants. Because of the low amount of the participants (N= 6,8 or15). The authors could have chosen to focus more on the qualitative data.

Validity of the findings

As mentioned above, maybe the conclusion is a little too firm. Although I don't doubt using the guidelines will positively support students’ learning, the low number of participants per course can not be used to generalise.

I do agree with the authors this paper and their guideline can fill a need for educators.

Comments for the author

Thanks for the contribution, love to read the revision.

Reviewer 3 ·

Basic reporting

The article provides a survey of empirical studies in general and with special emphasis for using in software engineering courses.
The article is written in English using clear and unambiguous text and conforms to professional standards. The article includes sufficient introduction and background in terms of related work. Comprehensive relevant literature is referenced.
Instead of figures, several tables with relevant content of the article are included.
The layout of the tables (border) should still be adapted.
Please note that I am not a native speaker so that I can not finally evaluate the written text. But I found the following typos:
- row 16: "necessity of"...?
- Table 2, P01, row 5: "Formulate use[r?]- or customer related.."
- row 539: "Especially the [the] practical projects .."
- row 540: "the workshops [were] highlighted."
Further minor problems are the following:
- row 55: A sentence about the content of Section 5 is missing.
- row 127: I miss a reference to SWEBOK.
- Table 1: I would like to have a third column with a description from the teachers perspective as describe in the text of the Subsections 3.1 to 3.6
- Table 4: Is "Priority" really the right term?
- Table 5 and 6 should be reconsidered. For example
-- Challenge C05 is also addressed in both tables and not as textually described more in Table 6 (row 447).
-- Why C04 is not relevant for case studies in both settings?

Experimental design

The experimental setting is described in Section 5.
The authors report about experiences with empirical studies in two software engineering courses. Their experiences are divided into a formal and an informal evaluation.
The number of the samples are rather small. The courses were attended at most by 19 students. It would be interesting if and how it is possible to perform empirical studies in courses with a large numer of students (100 and more).
In Example 2, the relationship of the terms "groups" and "teams" are not clearly explained. It is confusing to read.
References:
- row 632: I miss the url of the referenced portal.

Validity of the findings

The findings (Table 8/9/10/11 and the discussion) could be doubted because of the small sample. But from the point of view of an experienced teacher who also used empirical studies in SE education, the findings can be accepted/believed.
The selection criteria for the choice of a study (Subsection 4.5 and Table 7) is based on literature and "experience". But it needs more explanations. Furthermore I miss a legend: what do +/o/-/"o/+" and so on mean?

Comments for the author

I like the article especially because of the classification of empirical studies and their hints for the use in SE education in terms of purposes, challenges and validity considerations.
All in all the first part of the article seems to be more mature than the second part.

---

## Round 0.2 · Minor Revisions

· Academic Editor

Minor Revisions

Thank you for your response to the earlier reviews. I have decided for "minor revisions" despite the fact that one reviewer still recommends "major revision": I'm trusting you to act on the comments made by all three reviewers.

Reviewer 1 ·

Basic reporting

Well written

Experimental design

No comment

Validity of the findings

Fine

Comments for the author

I’d like to thank the authors for their constructive responses and extensive changes. Table 8 is particularly valuable and the example in 6.3.

There’s not much to add. I suppose we still have slight differences in what are the most valuable aspects; I think the examples and experiences are the most important whilst the authors also place emphasis upon setting out the underlying basics such as background on underlying empirical methods. However, I think the paper is considerably improved and can be published.

It would still be useful to label the first column P01 etc for Tables 2-4 i.e. student purpose etc as it’s not until l537 I can begin to infer what they mean.

·

Basic reporting

The authors wrote a clear and well structured paper about their experiences in applying empirical research methods in their software engineering (SE) courses. From literature they selected five research types: experiment, case study, continuous experimentation, simulation and individual. Based on their own experience the authors suggest guidelines for using empirical studies in SE education. They describe 3 cases in which they applied empirical studies in education. Based on the course evaluation data from their students the authors conclude their students are more motivated and get higher scores when using empirical studies in education.

A strength of the paper is that the suggested study types are all supported with examples of own (published) research.
Another strength is that the examples use an educational perspective and an industrial perspective.

A weakness of the paper is that they don’t discuss any validity threats. You would expect that from a paper that advertises research as an instrument.

- Title
Maybe the authors should go for ‘studies’ in stead of ‘instruments’. Although ‘instruments’ is used, the paper discusses ‘studies’. For the rest the title reflects the content of the paper.

- Introduction
line 19,20,21: rephrase, no it says that the goal of the paper is to develop guidelines. The authors probably mean to show that the already developed guidelines work in the observed cases.

- Related work
The related work describes application in other disciplines, requirements and application in SE education.
line: 125 ‘little attention …’ The authors are somewhat negative about the current application in SE education. In educational approaches such as problem based and project based learning students apply empirical instruments. Also case study approaches in collaboration with industry are widely used. It could be more though.

- Research approach
line 169: which papers are analysed and why were they important and leading?

- An Overview Of …
line 186: why are these instruments listed, on what ground did they author choose them? Why are they common?
Table 1: how was this table constructed, on what ground?
line 293: ‘our experience show .. requirements’ Is there source that claims this is often the case?
line 420: ‘from a teaching perspective’ -> ‘from an educational perspective’ (maybe also other places in the text)
line 437: ‘Further Instruments’ -> ‘Supporting Instruments’ ?

- A Guideline …
line 518: ‘we identified … ‘ -> how?
line 531: ‘we derived …’ -> from what?
Table 3: are there possible resource challenges in terms of amount of lecturers / hours?
Table 5/7: How were the tables created (process, arguments, etc.)?
line 571: ‘For students … consideration’ a credit point reflects time, it is a fixed number. How could different times be ‘required’ for a credit?
line 603: This subsection is very unclear. On what ground was it included? Why and how were own criteria added?

- Experiences
suggestion: include the study type in all the subsection titles
It is not clear on what ground the authors claim the basic course content is the same. Is it really comparable?
It is very confusing that in the text of example 1 and in table 9 the authors write about grades. It seems that it should be about student judgements. No exam grades are provided.
How many students were involved?
line 690: Example 3, this example is obvious. What is new about using empirical studies in the master thesis period? The example does not describe some special approach in comparison
to more conventional approaches.
line 757: is this meant as overal discussion? Make it a section then. Otherwise: you are missing a discussion section.
line 770: first use of ‘exams’

- Conclusion
line 775: ‘there is a lack of guidance’ I think there still is. The authors did provide compact guideline in the form of tables. They really can be of help for lecturers to setup their courses using empirical studies. The guidelines in this paper don’t provide guidance yet. Guidance suggests that there is a guide (handbook), which would be great actually.
line 780: ‘Our experience .. exams’ Here the authors claim, based on 1 comparison of grades (that are not show in the paper) the use of empirical studies have a positive influence.
Strongly advise: rephrase this, or show the evidence. Still it is only 1 comparison.
line 790/791: sentence does not ‘flow’.
line 798 - 804: maybe rephrase this. Are the authors mention matters twice? It is not clear what they mean here. What is the difference between ‘student assessment’ on line 798 and ‘assess learning outcomes’ on line 802?

Experimental design

The authors did not really conducted an experiment. They claim to have a mixed method research design, but this is not reflected in the paper. The qualitative part is very clear and well structured. The quantitative part is not well described. The authors present tables without explanation of the data collection process. Was it done in previous research? The authors do not discuss the participants (students, lecturers, other staff members) that participated in the studies. The authors don’t write in much detail about the data collection and data analysis processes.

Validity of the findings

The authors do not discuss any validity threats. From the paper it is not clear how the quantitative data was collected and how large the sample was. The authors mention exam grades, but it is not clear where in the paper this is discussed in more detail. Still the authors base a conclusion on that data. The conclusions based on the qualitative data is more and better discussed, but the authors should have taken the responsibility to discuss the validity of their research (for example biases, construct validity, conclusion validity etc.).

Comments for the author

I agree with the authors that this paper and the guideline its presents can be of great value for educators. I hope the authors will revise the paper. I like to see it published.

Reviewer 3 ·

Basic reporting

The article is written in English using clear and unambiguous text and conforms to professional standards of courtesy and expression.
However, the article should be finally reviewed by a native speaker.

The article includes a sufficient introduction and background to demonstrate how the work fits into the broader field of knowledge.
Relevant and comprehensive literature is appropriately referenced.
In the keyword list, "Onboarding" should be removed because this term is not used in the article (only in the title of a referenced article).

The structure of the article was improved in the first revision and now conforms to an acceptable format.
The article includes 12 tables which are relevant to the content. However, the layout of the tables (bordering) must be unified.
Furthermore, there is an inconsistency in Table 1: "Other instruments" is related to Subsection 4.6 renamed as "Further Instruments".
Aside from this inconsistency, the title "Instruments" is confusing because what you mean are the subjects of empirical study types
(Replication studies and OSS as crosscutting aspects).
I would also prefer to use the term "empirical study types" instead of "empirical instruments" in the title of the article.

The submission is "self-contained" that means it includes all relevant results with respect to the intention of the article.

Results basically include clear definitions of all terms and explanatory statements for the listed guidelines.
Minor issues are the following:
- Table 8: Change "Bachelor's level (first 4 years of study)" accordingly Row 611 to "Bachelor's level (0-3 years of study)"
- In Subsections 6.3 and 6.4, the authors write about types of case studies which are not listed in Section 4: Intrinsic case study, explorative deductive study, and qualitative survey design, as well as explorative (curiosity-driven) and confirmative experiments. These terms "break" the used framework of terms.

Experimental design

The submission clearly defines the research approach, which is relevant and meaningful for teachers in software engineering education.
The knowledge gap being investigated is identified. The guidelines for integrating empirical studies with software engineering courses contribute to filling that gap.
The investigation is performed to a high standard. The used research method is described with sufficient details.

Validity of the findings

The authors emphasize that the article provides an "INITIAL systematization of empirical instruments from the educational perspective" and also an INITIAL guideline based on their teaching experience.
They describe three examples of courses (Section 6) and discuss the limitations of their work.
The authors also clarify that further studies are needed to assess learning outcomes of courses using their proposed guidelines.

Comments for the author

I like your first revision of your article with a clearer structure and clarifications of open issues. I see your guidelines as a basis for further discussions and experiments in software engineering education.
I recommend the article to be accepted if you consider in some revisions my minor issues and comments.

---

## Round 0.3 · accepted · Accept

· Academic Editor

Accept

Thank you for acting on the review comments.